# Establishment of reference intervals of clinical chemistry analytes for the adult population in Egypt

Heba Baz[1], Kiyoshi Ichihara[2]*, May Selim[1], Ahmed Awad[3], Sarah Aglan[3], Dalia Ramadan[1], Amina Hassab[4], Lamia Mansour[1], Ola Elgaddar[3]

1 Chemical Pathology Department, Cairo University, Cairo, Egypt, 2 Faculty of Health Sciences, Department of Clinical Laboratory Sciences, Yamaguchi University Graduate School of Medicine, Ube, Japan, 3 Chemical Pathology Department, Medical Research Institute, Alexandria University, Alexandria, Egypt, 4 Clinical Pathology Department, Alexandria University, Alexandria, Egypt

* ichihara@yamaguchi-u.ac.jp

## Abstract

**Data Availability Statement:** All relevant data are within the paper and it's supporting information files.

### Background

This is the first Egyptian nationwide study for derivation of reference intervals (RIs) for 34 major chemistry analytes. It was conducted as a part of the global initiative by the IFCC Committee on Reference Intervals and Decision Limits (C-RIDL) for establishing country-specific RIs based on a harmonized protocol.

### Methods

691 apparently healthy volunteers aged ≥18 years were recruited from multiple regions in Egypt. Serum specimens were analyzed in two centers. The harmonization and standardization of test results were achieved by measuring value-assigned serum panel provided by C-RIDL. The RIs were calculated by parametric method. Sources of variation of reference values (RVs) were evaluated by multiple regression analysis. The need for partitioning by sex, age, and region was judged primarily by standard deviation ratio (SDR).

### Results

Gender-specific RIs were required for six analytes including total bilirubin (TBil), aspartate and alanine aminotransferase (AST, ALT). Seven analytes required age-partitioning including glucose and low-density lipoprotein cholesterol (LDL-C). Regional differences were observed between northern and southern Egypt for direct bilirubin, glucose, and high-density-lipoprotein cholesterol (HDL-C) with all their RVs lower in southern Egypt. Compared with other collaborating countries, the features of Egyptian RVs were lower HDL-C and TBil and higher TG and C-reactive protein. In addition, BMI showed weak association with most of nutritional markers. These features were shared with two other Middle Eastern countries: Saudi Arabia and Turkey.

**Funding:** The part of the study that took place at Cairo University has been funded by a Cairo University research fund awarded to HB and MS. The part fulfilled in Alexandria university as well as all the immunoturbidimetric assays of the whole study were funded by the Japan Society for the Promotion of Science (JSPS), Scientific Research Fund number (16H02771: 2016–2019) awarded to KI. None of the authors received any salaries from any commercial body, neither were there any sponsorship of any sort.

**Competing interests:** No authors have competing interests.

**Abbreviations:** aCa, adjusted calcium; Alb, albumin; ALP, alkaline phosphatase; ALT, alanine aminotransferase; AMY, amylase; ANOVA, analysis of variance; AST, aspartate aminotransferase; BMI, body mass index; BR, bias ratio; C3, complement component 3; C4, complement component 4; Ca, calcium; CI, confidence interval; CK, creatine kinase; Cl, chloride; Cre, creatinine; C-RIDL, committee on reference intervals and decision limits; CRP, C-reactive protein; DBil, direct bilirubin; eGFR, estimated glomerular filtration rate; Fe, iron; GGT, γ-glutamyl transferase; Glb, globulin; Glu, glucose; HDL-C, high-density lipoprotein cholesterol; IgA, immunoglobulin A; IgG, immunoglobulin G; IgM, immunoglobulin M; IP, inorganic phosphate; K, potassium; LAVE, latent abnormal values exclusion; LDH, lactate dehydrogenase; LDL-C, low-density lipoprotein cholesterol; LL, lower limit; Mg, magnesium; MRA, multiple regression analysis; Na, sodium; nonHDL, non-HDL-C; PTH, parathyroid hormone; RI, reference interval; rp, partial correlation coefficient; RV, reference value; SD, standard deviation; SDR, standard deviation ratio; SV, Source of variation; TBil, total bilirubin; TC, total cholesterol; TG, triglycerides; TIBC, total iron binding capacity; TP, total protein; UA, uric acid; UIBC, unsaturated iron binding capacity; UL, upper limit.

## Conclusion

The standardized RIs established by this study can be used as common Egyptian RI, except for a few analytes that showed regional differences. Despite high prevalence of obesity among Egyptians, their RVs of nutritional markers are less sensitive to increased BMI, compared to other collaborating countries.

## Introduction

Egypt, being at the crossroads of Africa and Asia overlooking the Mediterranean Sea, had created a demographic melting pot of ethnicities over a long period of history. Thus, the study of the reference intervals (RIs) of common chemistry analytes and their sources of variations (SVs) in Egyptians in comparison to other nations is intriguing.

This study comes as the first Egyptian initiative to report RIs for 34 major laboratory analytes: 27 chemistry analytes, 6 immunoturbidimetric analytes, and parathyroid hormone. In 2013, the Committee on Reference Intervals and Decision Limits (C-RIDL), International Federation of Clinical Chemistry (IFCC) issued a harmonized protocol for establishing regional RIs by means of multicenter study [1]. For standardization and harmonization of tests results, it provided a value-assigned serum panel to collaborating countries on request [2]. Besides determining the RIs for common analytes, this study aimed at investigating the sources of variation that may impact the RIs, including age, gender, body mass index (BMI) and region.

Egypt is the third Middle Eastern country following Saudi Arabia and Turkey that adopted the C-RIDL protocol to derive country-specific RIs for nationwide use. The results of this study provide an opportunity to compare the RIs among Middle Eastern countries, and across other collaborating countries of widely different demographic profiles such as body mass index (BMI), alcohol intake and smoking.

## Materials and methods

### 1. Study design

This study was conducted by a collaboration of Cairo University Faculty of Medicine and Medical Research Institute of Alexandria University. The study was approved by the ethical committee of Cairo University (N-13-2015) on January 31$^{st}$, 2015, and the final data analysis ended early 2020. Each participant signed an informed consent form before enrollment.

### 2. Recruitment of reference individuals

A total of 691 apparently healthy Egyptian volunteers were recruited from Cairo region [Cairo, Giza, Helwan], Alexandria, the Nile Delta [Beheyra, Daqahlia, Damietta, Gharbia, Kar Elshiekh, Menoufia], South (Upper) Egypt [Fayoum, Beni-Sueif, Minya, Assiut, Qena, and Nubia], and the Suez Canal [Port Said and Suez]. The subjects aged more than 18 years and were stratified into 5 age groups: 18–29, 30–39, 40–49, 50–65 and ≥65 years. Subjects were selected so that at least 80% were from the age of 18 to 65 years, with equal gender mix and age distributions, except for individuals over 65 years of age. We followed the inclusion and exclusion criteria of C-RIDL protocol: Participants of the study were included if they are feeling subjectively well, older than 18 years with no upper limit of age for participation. Those who take less equal 3 drugs for minor conditions or nutritional supplements were allowed. All the following were excluded: participants who were known diabetics on insulin or oral therapy,

who had a positive history of hepatic or renal diseases, who had an grossly abnormal test results in the previous year, who were hospitalized within the previous 4 weeks prior to participating in the study, who donated blood in the previous 3 months, known carrier for HCV, HBV or HIV, who participated recently in a clinical trial for an investigational product.

## 3. Blood sampling

The volunteers were instructed to avoid excessive exercise and eating a few days before the sampling. The time of sampling was set from 8–11 AM, after resting in a seated position for approximately 20 minutes [3]. In the resting period, each participant answered to the detailed health-status questionnaire that was adopted from the harmonized protocol. Ten mL of blood were drawn into a plain evacuated tube. Clotted samples were centrifuged, and the sera were divided and stored at –80˚C.

## 4. Analytes and measurements

The following 27 chemistry analytes were analyzed spectrophotometrically using dedicated manufacturer reagents: total protein (TP), albumin (Alb), uric acid (UA), urea, creatinine (Cre), calcium (Ca), inorganic phosphate (IP), magnesium (Mg), iron (Fe), unsaturated iron binding capacity (UIBC), total bilirubin (TBil), direct bilirubin (DBil), glucose (Glu), total cholesterol (TC), triglycerides (TG), high-density lipoprotein cholesterol (HDL-C), low-density lipoprotein cholesterol (LDL-C), aspartate transaminase (AST), alanine transaminase (ALT), lactate dehydrogenase (LDH), Alkaline Phosphatase (ALP), γ-glutamyl transferase (GGT), creatine kinase (CK), and amylase (AMY). Three analytes were assayed by Ion selective electrodes: sodium(Na), potassium(K), chloride (Cl). Six analytes were assayed by immunoturbidimetric methods namely: immunoglobulins G, A, and M (IgG, IgA, IgM), complement components 3 and 4 (C3, C4) and C-reactive protein (CRP), parathormone (PTH) was assayed by electro-chemiluminescent immunoassay. **S1 Table** shows details of assay methods and traceability of each measurement as well as formulae for calculating the following calculated parameters: globulin (Glb), estimated glomerular filtration rate (eGFR) [4], adjusted calcium (aCa), total iron binding capacity (TIBC), and non-HDL-C (nonHDL).

Measurements were performed in a batch of 60~150 specimens/per day, after thawing at room temperature for at least 1 hour. Chemistry samples collected by Cairo University were analyzed on Beckman AU 680 (Beckman Coulter International, Nyon, Switzerland), while chemistry samples collected by Alexandria University as well as all samples for immunoturbidimetry and PTH from the two centers were measured on Cobas 6000 modular system (Cobas modular Roche, Indianapolis IN, USA), Roche, at Hassab Labs in Alexandria. Both labs are ISO 15189 accredited. Samples were assayed within 2−3 months of storage at −80˚C except for the following analytes: IgG, IgM, IgA, C3, C4, CRP and PTH, which were measured 8 months after the collection.

## 5. Between-center method comparison and standardization

A panel composed of 50 healthy volunteers' sera with assigned values for 34 chemistry and immunoturbidimetry analytes [2,5] was measured by the two centers for merging and standardization of test results as described elsewhere For between-day quality control, "mini-panel" was prepared comprising sera of 4 volunteers measure at each run of assay in both centers [2].

## 6 Statistical analyses

**6.1. Sources of variation of reference values.** Multiple regression analysis (MRA) was performed, separately for males and females, to examine sources of variation. Reference values (RVs) of each analyte were set as an objective variable, while set as explanatory variables were volunteers' region, age, body mass index (BMI), levels of cigarette smoking, exercise, consumption of soft drinks. For the region, we categorized the volunteers by their origin either as northern or southern (Upper) Egypt, where northern Egypt included inhabitants of Alexandria, Delta, Cairo, and Suez Canal, while southern Egypt included inhabitants from Fayoum, Beni-Sueif or areas southern as far as Nubia. Northern Egypt was set as the reference category: [northern] = 0, [southern] = 1. The degree of association of each explanatory variable with the objective variable was expressed as a standardized partial regression coefficient (rp), which generally takes a value between −1.0 and 1.0. We regarded |rp|≥0.2 as an appreciable effect size of the association between small correlation (0.1) and medium correlation (0.3), specified by Cohen [6].

**6.2. Partitioning criteria.** To judge the need for partitioning RVs by sex, age-subgroup (<45 versus ≥45 years), and region (northern or southern Egypt), we primarily used standard deviation ratio (SDR) based on analysis of variance (ANOVA). The partitioning of RVs by age was done arbitrarily at 45 years to ensure enough data size for the higher age group. SDR for a particular SV (SDRsv) was calculated as standard deviation (SD) due to a particular source of variation (SDsv) divided by SD due to "coarse" between-individual SD (SDbi) or SD comprising the RI($SD_{RI}$). We regarded SDRs ≥ 0.3 as an appreciable level of between-subgroup variations, and SDRs≥0.4 as requiring partitioning of RVs by the factor concerned [7].

Because SDR represents the magnitude of between-subgroup differences at the central part of RV distribution, it may not reflect between-subgroup differences at lower or upper limits (LL or UL) of the RI. Therefore, as a secondary index of between-subgroup difference, "bias ratio" (BR) was computed at LL and UL. For example, the equation shown below for calculating BR of between-sex difference at UL ($BR_{UL}$),

$$BR_{UL} = \frac{UL_M - UL_F}{(UL_{MF} - LL_{MF})/3.92}$$

where subscript M, F, and MF represent male, female, and male + female, respectively, the same for the calculation of bias ratio for LL ($BR_{LL}$).

According to the conventional specification of allowable bias at the minimum level: $0.375 \times \sqrt{SD_G^2 + SD_I^2} = (SD_{RI})$ [8] where $SD_G$ and $SD_I$ represent "pure" between- and within-individual variations, we regard $BR_{UL}$ (or $BR_{LL}$) > 0.375 as a secondary criterion for partitioning RVs by sex, age, and region.

In both MRA and ANOVA, the RVs of analytes that showed highly skewed distributions were transformed logarithmically before the analyses to avoid excessive influence of deviated data in the periphery of RV distributions. Such analytes included TG, AST, ALT, LDH, GGT, CK, and CRP were performed using a statistical software, StatFlex Ver.7 (Artech, Osaka, Japan).

**6.3. Derivation of reference intervals.** The parametric method was used for computing reference intervals after transforming the distribution of RVs into Gaussian form using the modified Box-Cox transformation [7] to obtain mean and SD. The RI was calculated as the mean±1.96SD, which corresponds to the central 95% limits or LL and UL under transformed scale, then they were reverse-transformed to get the LL and UL on the original scale.

For improved precision of the RIs, the bootstrap method through 50-times resampling of the dataset was applied to obtain smoothed lower and upper limits (LL, UL), and mean of the

reference intervals. This resampling procedure was also used for predicting 90% confidence intervals (CI) for the limits of the reference interval.

The LAVE method [7] was applied for the secondary exclusion of individuals with conditions affecting nutritional, muscular, and inflammatory markers. In our study, the LAVE method was applied in two groups using a different set of reference tests. Group 1 consisted of nutritional, muscular markers, enzymes: Alb, UA, Glu, TC, TG, HDL-C, LDL-C, nonHDL, AST, ALT, LDH, GGT, CK, AMY, CRP. Group 2 consisted of TP, Alb, Glb, Na, K, Cl, Ca, Fe, UIBC, TIBC, IgG, IgA, IgM, C3, C4, and CRP. The reference tests used for LAVE for each group were 11 analytes for Group 1 (Alb, UA, Glu, TG, nonHDL-C, AST, ALT, LDH, GGT, CK, CRP), and 10 analytes for Group 2: Alb, Glb, Na, Cl, Ca, Fe, UIBC, IgG, C3, CRP.

## Results

### 1. Demographic profile of participants (S2 Table)

The 691 participants in our study included 323 males and 368 females. Seven individuals were excluded due to overt diseases: infection with high CRP, extreme hyperlipidemia, hepatitis C antibody positivity. Mean±SD of BMI for Egyptians was 28.9±5.5 kg/m$^2$ in females and 27.5 ±4.4 kg/m$^2$ in males. Females were found to have a slightly increasing BMI with advancing age while males remained almost the same. Regions of origin of participants were Southern Egypt (Upper Egypt) 38.5% (including Nubia accounting for 0.4%), and Northern Egypt 61.5% (Cairo 30%, Delta and Suez Canal 17.5%, Alexandria 14%). As for physical exercise, only 2% reported regular exercise from one to six days a week. Participants with smoking habit were 27.5% in males, and 1.1% in females and those who did not deny drinking alcohol were 1.25% in males and 0.6% in females.

From the health status questionnaire, we found regular use of drugs in the following frequencies: anti-hypertensives in 13 volunteers (1.9%), analgesics 51 (7.5%), anti-allergic 10 (1.6%), antacid 24 (3.5%), statin 4 (0.6%), others 8 (1.2%). We disregard the information in the subsequent analyses because they were all minor and small in dosage.

### 2. Merging the datasets from the two participating laboratories

For the analytes with assigned values in the serum panel, (i.e. the standardized analytes), the test results of the serum panel measured in common in the two centers were used to compare the two participating laboratories (S1 Fig shows the comparison between the panel results from the two laboratories). Among these analytes, obvious dissociation of panel results was observed between the two centers for Alb, Cre, Na, Cl, and Mg, and slight dissociations for HDL-C, AST, ALP, and GGT. This warranted recalibration of the results of each center independently to the assigned values for merging by use of major-axis linear regression lines. Then the results of the two centers were merged into one data set.

For the non-standardized analytes: TP, TBil, DBil, TIBC, UIBC; the test results from Alexandria University were aligned to those of Cairo University by use of the major axis regression lines constructed from the panel test results of the two centers.

### 3. Sources of variation and partitioning of reference values

Results of MRA are shown in Table 1 by setting a threshold of practical significance (effect size) as rp = 0.2. Age, BMI, and regionality (North vs. South Egypt) were the three major SVs. In males, with the advancement of age, urea, Cre, aCa, and TG increased, while eGFR and ALT decreased. Increasing BMI was associated with an increase in nutritional markers like UA, TC, LDL-C, nonHDL, C3, and C4, but the level of association was slight with their rp

**Table 1. Multiple regression analysis for source of variations of RVs.**

**Male**

| Obj var | n | R | Age | BMI | Region | Smoke |
|---|---|---|---|---|---|---|
| TP | 316 | 0.113 | -0.005 | 0.085 | -0.035 | -0.070 |
| Alb | 314 | 0.191 | -0.180 | 0.006 | 0.043 | 0.010 |
| Glb | 315 | 0.244 | 0.140 | 0.101 | -0.095 | -0.119 |
| UA | 315 | 0.277 | 0.159 | **0.205** | -0.025 | -0.011 |
| Urea | 313 | 0.245 | **0.234** | -0.042 | -0.035 | -0.054 |
| Cre | 313 | 0.275 | **0.221** | -0.004 | -0.134 | 0.009 |
| eGFR | 315 | 0.703 | **-0.686** | 0.005 | 0.083 | 0.015 |
| Na | 276 | 0.139 | 0.120 | 0.042 | 0.057 | 0.030 |
| K | 274 | 0.139 | 0.074 | 0.103 | 0.001 | -0.031 |
| Cl | 275 | 0.058 | 0.009 | -0.021 | 0.021 | 0.048 |
| Ca | 315 | 0.091 | 0.051 | 0.025 | 0.072 | -0.024 |
| aCa | 316 | 0.211 | **0.209** | 0.017 | 0.029 | -0.026 |
| IP | 314 | 0.117 | 0.031 | -0.007 | -0.105 | 0.033 |
| Mg | 315 | 0.094 | 0.024 | -0.013 | -0.038 | 0.082 |
| Fe | 315 | 0.121 | -0.003 | -0.050 | -0.046 | 0.100 |
| TIBC | 314 | 0.171 | -0.150 | 0.057 | 0.029 | -0.053 |
| UIBC | 315 | 0.198 | -0.145 | 0.059 | 0.069 | -0.090 |
| TBil | 309 | 0.213 | -0.006 | -0.139 | -0.140 | 0.054 |
| DBil | 316 | 0.275 | -0.115 | -0.066 | **-0.235** | 0.101 |
| Glu | 307 | 0.230 | 0.111 | 0.164 | -0.094 | 0.004 |
| TC | 316 | 0.233 | 0.041 | **0.217** | -0.012 | -0.058 |
| TG | 316 | 0.323 | **0.236** | 0.137 | 0.168 | -0.011 |
| HDL-C | 316 | 0.142 | -0.124 | 0.000 | -0.022 | -0.067 |
| LDL-C | 316 | 0.251 | 0.022 | **0.234** | -0.084 | -0.054 |
| nonHDL | 315 | 0.254 | 0.070 | **0.233** | -0.001 | -0.034 |
| AST | 313 | 0.144 | 0.036 | 0.083 | 0.066 | 0.079 |
| ALT | 316 | 0.326 | -0.257 | 0.160 | 0.100 | 0.065 |
| LDH | 310 | 0.175 | -0.053 | 0.046 | 0.149 | -0.010 |
| ALP | 312 | 0.198 | 0.162 | 0.029 | 0.004 | 0.101 |
| GGT | 315 | 0.240 | 0.193 | 0.085 | -0.047 | 0.064 |
| CK | 309 | 0.162 | -0.106 | 0.104 | -0.027 | 0.094 |
| AMY | 315 | 0.223 | 0.197 | -0.144 | 0.034 | 0.002 |
| IgG | 239 | 0.211 | -0.012 | 0.060 | 0.131 | -0.158 |
| IgA | 246 | 0.250 | 0.085 | 0.092 | 0.054 | **-0.211** |
| IgM | 251 | 0.210 | -0.181 | -0.012 | 0.056 | 0.074 |
| C3 | 249 | 0.270 | 0.030 | **0.264** | -0.012 | -0.030 |
| C4 | 250 | 0.259 | 0.099 | **0.206** | -0.025 | -0.098 |
| CRP | 248 | 0.228 | 0.099 | 0.190 | 0.011 | -0.006 |
| PTH | 249 | 0.132 | 0.100 | 0.067 | 0.031 | 0.009 |

**Female**

| Obj var | n | R | Age | BMI | Region |
|---|---|---|---|---|---|
| TP | 356 | 0.090 | 0.053 | 0.044 | -0.039 |
| Alb | 357 | 0.136 | 0.050 | -0.139 | -0.017 |
| Glb | 355 | 0.201 | 0.013 | 0.188 | -0.058 |
| UA | 359 | 0.307 | 0.190 | 0.168 | -0.088 |
| Urea | 358 | 0.242 | **0.227** | -0.088 | -0.076 |
| Cre | 358 | 0.263 | **0.221** | 0.009 | -0.112 |
| eGFR | 358 | 0.781 | **-0.775** | 0.000 | 0.037 |
| Na | 317 | 0.223 | -0.079 | **0.206** | 0.081 |
| K | 318 | 0.273 | 0.087 | 0.198 | 0.153 |
| Cl | 314 | 0.146 | -0.137 | 0.095 | -0.005 |
| Ca | 355 | 0.086 | 0.049 | 0.035 | -0.046 |
| aCa | 354 | 0.155 | 0.003 | 0.145 | -0.051 |
| IP | 359 | 0.144 | 0.069 | -0.027 | -0.118 |
| Mg | 357 | 0.133 | 0.116 | -0.017 | -0.056 |
| Fe | 358 | 0.186 | -0.010 | 0.040 | -0.184 |
| TIBC | 358 | 0.108 | -0.074 | 0.050 | -0.087 |
| UIBC | 359 | 0.040 | -0.038 | 0.021 | 0.008 |
| TBil | 358 | 0.174 | -0.028 | 0.008 | -0.176 |
| DBil | 356 | 0.367 | -0.182 | 0.039 | **-0.348** |
| Glu | 352 | 0.370 | 0.089 | 0.156 | **-0.301** |
| TC | 358 | 0.437 | **0.267** | **0.204** | -0.186 |
| TG | 359 | 0.415 | **0.292** | 0.182 | 0.198 |
| HDL-C | 358 | 0.309 | -0.053 | 0.025 | **-0.312** |
| LDL-C | 358 | 0.407 | **0.279** | 0.139 | -0.180 |
| nonHDL | 357 | 0.448 | **0.349** | 0.176 | -0.073 |
| AST | 358 | 0.156 | 0.066 | 0.104 | 0.082 |
| ALT | 359 | 0.131 | -0.101 | 0.104 | -0.060 |
| LDH | 357 | 0.289 | 0.046 | 0.170 | **0.225** |
| ALP | 359 | 0.295 | **0.245** | 0.106 | -0.013 |
| GGT | 357 | 0.362 | **0.213** | 0.191 | -0.138 |
| CK | 359 | 0.124 | -0.036 | 0.129 | -0.017 |
| AMY | 357 | 0.144 | 0.010 | -0.146 | 0.006 |
| IgG | 324 | 0.175 | -0.073 | 0.095 | 0.136 |
| IgA | 326 | 0.175 | 0.142 | 0.072 | 0.019 |
| IgM | 326 | 0.165 | -0.147 | -0.027 | -0.079 |
| C3 | 330 | 0.221 | 0.087 | 0.179 | 0.054 |
| C4 | 328 | 0.203 | 0.079 | 0.088 | -0.139 |
| CRP | 328 | 0.337 | 0.128 | **0.268** | 0.123 |
| PTH | 326 | 0.082 | 0.045 | 0.039 | -0.040 |

Multiple regression analysis was performed separately for each sex by setting RVs of each analyte as objective variable and fixed set of explanatory variables. Listed in the table are standardized partial regression coefficient (rp). |rp| ≥ 0.20 was considered significant and shown in bold letter. The magnitude of negative |rp| was expressed by red bar and positive by green bar.

values between 0.20 and 0.26. It is of note that the associations of BMI with Glu, TG, AST, ALT, and CRP, major nutritional markers, were at rp<0.2. The regionality affected DBil with a rp value of −0.24, indicating lower DBil values among individuals from southern Egypt. Smoking habit showed a weak negative association with IgA: i.e., RVs of IgA were lower in those who smoked.

In females, age was positively associated with non-HDL, TG, LDL-C, TC, ALP, Urea, Cre, and GGT in this order of strength and negatively associated with eGFR. The association of BMI with nutritional markers was weak with |rp|<0.20 except for CRP, TC, and Na. As regards the regionality, RVs of southern Egypt were found lower (rp≤−0.2) for DBil, HDL-C, and Glu.

Further analyses of the SVs were performed using nested ANOVAs (**Table 2**), where the values of SDR was marked in two grades: SDR ≥0.3 in bold and SDR ≥0.4 in orange background, respectively considered as between-subgroup differences of "non-negligible" and "significant" degree.

Between-sex differences with significant SDRsex (shown in the parenthesis) were observed for Cre (0.97), UA (0.64), GGT (0.48), TBil (0.47), AST (0.41), CK (0.40) in the order of its magnitude. RVs of all the above analytes were higher in males than females. It is notable that SDRsex did not exceed the threshold of 0.4 in HDL-C (0.27), IgM (0.20), Alb (0.17), TG (0.11), and CRP (0.00), RIs of which are often partitioned by sex.

Significant between-age differences for nonHDL (0.50) among females was noted affirming the already noted significant rp values derived by MRA, as well as an expected significant SDRage for eGFR for both sexes. Graphical representation of sex and age-related changes for selected parameters are shown in **Fig 1**. **S2 Fig** shows all parameters.

Regarding the between-region difference (SDRreg), it was observed only in females for DBil (0.52), Glu (0.43), and HDL-C (0.42), with all of their RVs lower in the South compared to the North as shown in **S3 Fig**.

## 4. Derivation of reference intervals

RIs were derived by the parametric method with or without the LAVE. The calculations were done in two steps, first partitioning of RVs by sex, followed by further partitioning by age at 45 years (<45 versus ≥45 years). The RIs by the former calculation are listed for all analytes in **S3 Table** and those by the latter calculation are listed in **S4 Table** only for those analytes showing non-negligible levels of SDRage or BR$_{LL}$ or BR$_{UL}$.

From SDRsex>0.4, sex specific RIs were adopted for UA, Cre, TBil, AST, and GGT. On the other hand, although SDRsex was below 0.4, RIs were partitioned by sex due to high BR$_{LL}$ or BR$_{UL}$ (>0.375) for the following analytes: urea, eGFR, TIBC, UIBC, DBil, HDL-C, nonHDL, ALT, ALP, CK, AMY, IgG, and CRP. The exception to this decision scheme was applied to Alb, K, IP, Mg, Fe, and IgM because actual between-sex differences at LL or UL were small or 2~4 times of the reporting unit of each analyte.

Based on SDRage, RIs partitioned by age were found necessary only for eGFR (males and females) and nonHDL (female). While, based on significant BR$_{LL}$ or BR$_{UL}$ for between-age subgroup differences, age-partitioned RIs were also adopted in females for UA, TG, ALT, GGT, and CK, and in both sexes (without gender distinction) for Glu, TC, and LDL-C.

As for the need for applying the LAVE method, the decision was made based on BR$_{LL}$ or BR$_{UL}$ for the difference of RIs with/without LAVE. It turned out that the LAVE method was found effective in lowering the UL of the RIs in male for UA, TIBC, nonHDL, AST, TC, GGT, IgG, and CRP, and in female for TC, TG, AST, ALT, GGT, and CK. The rationale behind choosing each RI are shown in **S3 and S4 Tables.** The list of RIs thus chosen are **shown in Table 3**.

**Table 2. Standard deviation ratios (SDR) representing the magnitude of between-sex, between-region, and between-age variations of RVs.**

| Test item | SDRsex | SDRage | | SDRreg | |
|---|---|---|---|---|---|
| | | M | F | M | F |
| TP | 0.000 | 0.000 | 0.044 | 0.189 | 0.000 |
| Alb | 0.166 | 0.113 | 0.057 | 0.080 | 0.000 |
| Glb | 0.078 | 0.053 | 0.000 | 0.173 | 0.162 |
| UA | **0.635** | 0.184 | 0.265 | 0.109 | 0.044 |
| Urea* | **0.353** | **0.328** | 0.269 | 0.000 | 0.000 |
| Cre | **0.974** | 0.213 | 0.206 | 0.247 | 0.102 |
| eGFR | 0.000 | **1.025** | **1.147** | 0.296 | 0.000 |
| Na | 0.000 | 0.000 | 0.000 | 0.208 | 0.232 |
| K | 0.211 | 0.065 | 0.045 | 0.000 | 0.224 |
| Cl | 0.030 | 0.000 | 0.129 | 0.000 | 0.000 |
| Ca | 0.206 | 0.000 | 0.098 | 0.110 | 0.000 |
| aCa | 0.000 | 0.228 | 0.000 | 0.000 | 0.000 |
| IP | 0.000 | 0.000 | 0.198 | 0.166 | 0.244 |
| Mg | 0.189 | 0.000 | 0.096 | 0.000 | 0.099 |
| Fe* | **0.306** | 0.000 | 0.000 | 0.216 | 0.251 |
| TIBC | 0.266 | 0.205 | 0.000 | 0.000 | 0.133 |
| UIBC | **0.375** | 0.191 | 0.000 | 0.000 | 0.000 |
| TBil* | **0.471** | 0.000 | 0.000 | 0.185 | 0.241 |
| DBil* | **0.364** | 0.000 | 0.000 | **0.364** | **0.519** |
| Glu* | 0.000 | 0.109 | 0.000 | 0.090 | **0.428** |
| TC | 0.000 | 0.000 | **0.390** | 0.000 | 0.226 |
| TG* | 0.114 | 0.257 | 0.257 | 0.215 | **0.336** |
| HDL-C | 0.269 | 0.163 | 0.000 | 0.000 | **0.423** |
| LDL-C | 0.000 | 0.000 | **0.373** | 0.000 | 0.258 |
| nonHDL | 0.000 | 0.140 | **0.497** | 0.000 | 0.149 |
| AST* | **0.414** | 0.043 | 0.099 | 0.142 | 0.000 |
| ALT* | 0.292 | **0.373** | 0.000 | 0.223 | 0.138 |
| LDH* | 0.000 | 0.000 | 0.000 | 0.252 | **0.358** |
| ALP* | 0.231 | 0.228 | **0.386** | 0.000 | 0.000 |
| GGT* | **0.479** | **0.372** | **0.314** | 0.000 | 0.111 |
| CK* | **0.395** | 0.015 | 0.000 | 0.000 | 0.000 |
| AMY* | 0.244 | 0.241 | 0.000 | 0.081 | 0.000 |
| IgG | 0.159 | 0.000 | 0.000 | 0.110 | 0.215 |
| IgA* | 0.000 | 0.030 | 0.146 | 0.000 | 0.092 |
| IgM* | 0.199 | 0.173 | 0.058 | 0.120 | 0.106 |
| C3 | 0.061 | 0.000 | 0.207 | 0.115 | 0.000 |
| C4 | 0.000 | 0.119 | 0.000 | 0.107 | 0.181 |
| CRP* | 0.000 | 0.106 | 0.123 | 0.179 | 0.113 |
| PTH* | 0.000 | 0.131 | 0.016 | 0.000 | 0.071 |

## 5. Comparison of Egyptian RVs with other populations

Fig 2 shows between-ethnic group comparison of RVs for four analytes that we identified as peculiar to Egyptian population from what reported in the C-RIDL's global study [8,11].

They were RVs for TBil, TG, HDL-C, and CRP. Comparison were made among RVs of 10 countries collaborated in the global study. They were subgrouped by ethnicity as Middle Easterners: "Egypt, Saudi Arabia, Turkey", Asians: "Japan, China, India", and Caucasians "USA,

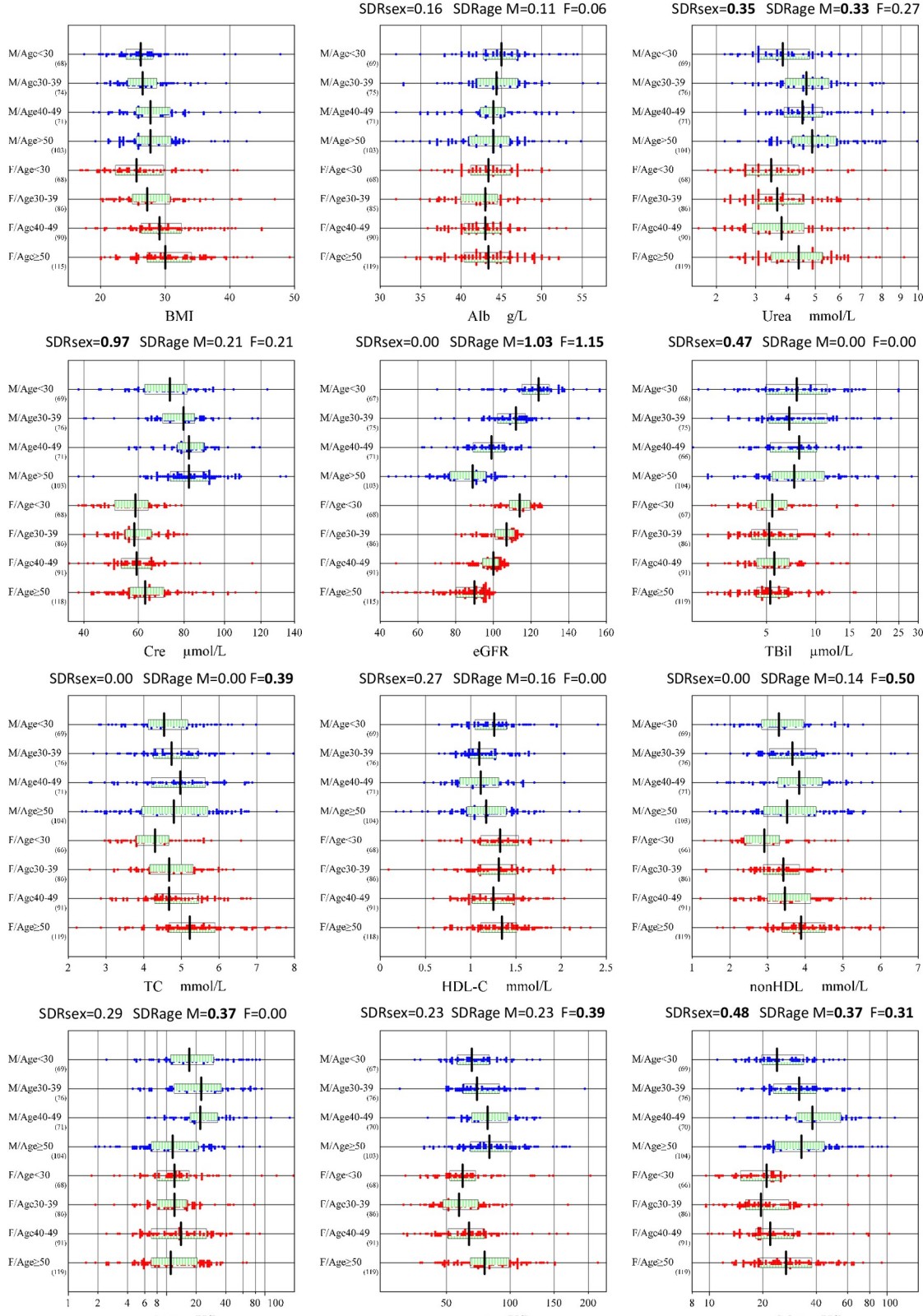

**Fig 1. Sex and age-related changes of selected parameters.** RVs were partitioned by sex (male: M, female: F) and age-subgroups (~29, 30~39, 40~49, 50~). The box in the center of each scattergram indicates the mid 50% range of RVs, and its central vertical bar

represents the median. The data size is shown at the right bottom of the age group labels. Because no secondary exclusion was done for RVs, the range of the scatter plot may not match to the RI to be determined.

Russia, Argentina, South African Caucasians (ZA-Cau)". It is notable that Egyptian RVs for TBil and HDL-C are shifted to the lower side, while those for TG and CRP are shifted upwards. These trends are generally similar among Middle Easterners.

## 6. Comparison of Egyptian BMI-related changes with other populations

To further explore how consistent the BMI related changes were across the ten countries that collaborated in the C-RIDL global study, **Fig 3** was drawn for six nutritional markers namely LDL-C, HDL-C, ALT, GGT, CRP, and UA.

## Discussion

Mean±SD of BMI in our study was $28.2 \pm 5.0$ kg/m$^2$. It is higher than those reported by most collaborating countries adopting the IFCC C-RIDL protocol namely [9]: Russia (26.6±4.5), South African Caucasians (25.9±3.7), Japan (22.9±2.6), China (23.6±3.0), Philippines (23±3.8), Saudi Arabia (28.5±5.6), Turkey (26.6±3.6), and India (males: 24.6 ± 3.46, females: 24.5 ± 4.4) [10]. Relevant to the obesity is the sedentary lifestyle reported by 98% of our participants, leaving only 2% exercising regularly at a rate of one to six days a week. The high BMI in comparison to other countries poses an interesting question regarding the effect of BMI on the derived RI.

The interim report by C-RIDL on the global multicenter study [5,9] highlighted close association of BMI in terms of rp with the following analytes: C3, ALT, CRP, UA, GGT, HDL-C, TG, LDL-C, C4, and AST in that order of strength in males, and similarly in females. In this study, we observed a similar association of BMI with those nutritional/inflammatory markers, but at much weaker degree: i.e., values of |rp| for ALT, CRP, GGT, and TG were all below 0.2.

For TC, UL of Egyptians was 6.28 mmol/L for both sexes <45 years, comparable to other studies: Turkey 6.2 [11], India 6.2 [10], Saudi Arabia 6.36 [12] and China 6.16 mmol/L [13], while among individuals >45 years, the Egyptian UL (6.99 mmol/L) was the highest reported followed by Indian UL of 6.7 (M) and 6.6 (F) mmol/L. Applying the LAVE procedure to TC was effective in lowering the UL for <45 years of age, but not for UL of >45 years. This failure of LAVE was previously encountered in the Indian study [10] and was attributed then to a weaker association among reference test items used for the LAVE procedure. With regard to HDL-C, **Fig 2** shows a tendency of Egyptians to have the lowest LL of 0.68 (M) and 0.72 (F) mmol/L followed by India that reported an LL of 0.70 (M) and 0.80 (F) mmol/L. Egyptian TG showed higher ULs of 4.06 (M), 2.72 (F:<45 years), 3.52 (F: ≥45) mmol/L than the ULs reported from Turkey 3.39 (M), 2.52 (F), and Saudi Arabia 3.58 (M), 1.60 (F) mmol/L. The contrast is even stronger if compared with Asian and Caucasian.

RIs for CRP RI in this study was among the highest reported with UL of 25.2 (M) and 35.3 (F) mg/L. This female predominance of high CRP was previously reported [10,12]. The UL of C4(0.50g/L) was comparable to both the manufacturer's UL (0.40 g/L) and the previously reported UL for Indian of 0.55 g/L [10]. However, C3 UL was obviously higher (2.38 g/L) compared to both the manufacturer's UL (1.8g/L) and previously reported UL by the latter study 1.82g/L. C3 is an acute phase protein. Increased levels may indicate a low-grade inflammatory response seen among inhabitants of equatorial and subequatorial terrains [14] or an underlying inflammation related to metabolic syndrome [15].

**Table 3. The list of RIs stratified by sex and age group.**

| Item | Unit | LAVE | Sex | Age | n | 90%CI of LL | | RI in SI unit | | | 90%CI of UL | | Unit | RI in conv. unit | | |
|---|---|---|---|---|---|---|---|---|---|---|---|---|---|---|---|---|
| | | | | | | L | H | LL | 1 | UL | L | H | | LL | Me | UL |
| TP | g/L | (-) | MF | 18~65 | 620 | 58.5 | 60.8 | 60 | 71 | 82 | 80.9 | 82.8 | g/L | 6.0 | 7.1 | 8.2 |
| Alb | g/L | (-) | MF | 18~65 | 620 | 34.9 | 36.3 | 36 | 44 | 51 | 50.3 | 51.8 | g/L | 3.6 | 4.4 | 5.1 |
| Glb | g/L | (-) | MF | 18~65 | 619 | 18.9 | 20.1 | 20 | 27 | 37 | 35.9 | 38.0 | g/L | 2.0 | 2.7 | 3.7 |
| UA | µmol/L | (-) | M | 18~65 | 289 | 149 | 188 | 168 | 326 | 479 | 463 | 495 | mg/dL | 2.8 | 5.4 | 7.9 |
| | | (-) | F | ~45 | 197 | 131 | 158 | 145 | 244 | 358 | 339 | 377 | | 2.4 | 4.0 | 5.8 |
| | | (-) | F | 45~ | 166 | 140 | 184 | 162 | 273 | 443 | 399 | 487 | | 2.6 | 4.4 | 7.2 |
| Urea | mmol/L | (-) | M | 18~65 | 288 | 2.50 | 2.81 | 2.7 | 4.5 | 7.6 | 7.2 | 8.0 | mg/dL | 16.2 | 27.0 | 45.6 |
| | | (-) | F | 18~65 | 334 | 1.99 | 2.25 | 2.1 | 3.7 | 7.0 | 6.7 | 7.3 | | 12.6 | 22.2 | 42.0 |
| Cre | µmol/L | (-) | M | 18~65 | 287 | 43 | 52 | 47 | 79 | 105 | 101 | 110 | mg/dL | 0.54 | 0.90 | 1.19 |
| | | (-) | F | 18~65 | 333 | 39 | 43 | 41 | 60 | 80 | 77 | 82 | | 0.46 | 0.68 | 0.90 |
| eGFR | ml/min/ 1.73m² | (-) | M | ~45 | 174 | 73 | 85 | 79 | 113 | 144 | 139 | 150 | ml/min/ 1.73m² | 79 | 113 | 144 |
| | | (-) | M | 45~ | 141 | 52 | 63 | 58 | 89 | 113 | 108 | 119 | | 58 | 89 | 113 |
| | | (-) | F | ~45 | 196 | 84 | 90 | 87 | 107 | 123 | 121 | 125 | | 87 | 107 | 123 |
| | | (-) | F | 45~ | 163 | 55 | 68 | 61 | 90 | 105 | 104 | 107 | | 61 | 90 | 105 |
| Na | mmol/L | (-) | MF | 18~65 | 544 | 130.1 | 131.5 | 131 | 139 | 148 | 147.0 | 148.7 | mmol/L | 131 | 140 | 148 |
| K | mmol/L | (-) | MF | 18~65 | 543 | 3.27 | 3.45 | 3.4 | 4.3 | 5.3 | 5.24 | 5.43 | mmol/L | 3.4 | 4.3 | 5.3 |
| Cl | mmol/L | (-) | MF | 18~65 | 538 | 93.6 | 95.0 | 94 | 104 | 111 | 110.3 | 111.9 | mmol/L | 94 | 104 | 111 |
| Ca | mmol/L | (-) | MF | 18~65 | 619 | 2.14 | 2.19 | 2.17 | 2.41 | 2.68 | 2.66 | 2.71 | mg/dL | 8.7 | 9.6 | 10.7 |
| aCa | mmol/L | (-) | MF | 18~65 | 618 | 2.04 | 2.11 | 2.08 | 2.33 | 2.54 | 2.52 | 2.57 | mg/dL | 8.3 | 9.3 | 10.2 |
| IP | mmol/L | (-) | MF | 18~65 | 624 | 0.77 | 0.81 | 0.79 | 1.11 | 1.52 | 1.48 | 1.55 | mg/dL | 2.4 | 3.4 | 4.7 |
| Mg | mmol/L | (-) | MF | 18~65 | 620 | 0.63 | 0.66 | 0.65 | 0.83 | 1.05 | 1.03 | 1.07 | mg/dL | 1.6 | 2.0 | 2.6 |
| Fe | µmol/L | (-) | MF | 18~65 | 618 | 3.1 | 3.9 | 3 | 12 | 26 | 25 | 27 | µg/dL | 19 | 65 | 145 |
| TIBC | µmol/L | (+) | M | 18~65 | 245 | 38 | 42 | 40 | 57 | 78 | 75 | 81 | µg/dL | 224 | 319 | 437 |
| | | (-) | F | 18~65 | 333 | 42 | 46 | 44 | 61 | 85 | 82 | 88 | | 246 | 341 | 475 |
| UIBC | µmol/L | (-) | M | 18~65 | 287 | 20 | 26 | 23 | 42 | 73 | 69 | 77 | µg/dL | 129 | 235 | 408 |
| | | (-) | F | 18~65 | 333 | 27 | 32 | 30 | 49 | 80 | 76 | 84 | | 165 | 273 | 447 |
| TBil | mmol/L | (-) | M | 18~65 | 282 | 2.1 | 3.0 | 2.5 | 7.6 | 20.6 | 18.1 | 23.1 | µg/dL | 0.15 | 0.45 | 1.20 |
| | | (-) | F | 18~65 | 332 | 2.0 | 2.5 | 2.2 | 5.4 | 12.4 | 11.1 | 13.7 | | 0.13 | 0.31 | 0.72 |
| DBil | mmol/L | (-) | M | 18~65 | 289 | 0.52 | 0.72 | 0.62 | 2.16 | 6.31 | 5.56 | 7.06 | µg/dL | 0.04 | 0.13 | 0.37 |
| | | (-) | F | 18~65 | 327 | 0.37 | 0.53 | 0.45 | 1.56 | 3.65 | 3.21 | 4.09 | | 0.03 | 0.09 | 0.21 |
| Glu | mmol/L | (+) | MF | ~45 | 277 | 2.96 | 3.35 | 3.15 | 4.61 | 6.19 | 5.94 | 6.44 | mg/dL | 56 | 83 | 111 |
| | | (-) | MF | 45~ | 301 | 3.00 | 3.43 | 3.22 | 5.04 | 7.10 | 6.64 | 7.56 | | 58 | 91 | 128 |
| TC | mmol/L | (+) | MF | ~45 | 276 | 3.41 | 3.55 | 3.48 | 4.61 | 6.28 | 6.11 | 6.46 | mg/dL | 135 | 178 | 243 |
| | | (+) | MF | 45~ | 250 | 3.27 | 3.57 | 3.42 | 5.10 | 6.99 | 6.77 | 7.21 | | 132 | 197 | 271 |
| TG | mmol/L | (-) | M | 18~65 | 286 | 0.43 | 0.56 | 0.50 | 1.27 | 4.06 | 3.63 | 4.48 | mg/dL | 44 | 112 | 359 |
| | | (-) | F | ~45 | 197 | 0.36 | 0.44 | 0.40 | 0.92 | 2.72 | 2.40 | 3.03 | | 36 | 82 | 240 |
| | | (+) | F | 45~ | 125 | 0.45 | 0.70 | 0.57 | 1.28 | 3.52 | 2.87 | 4.17 | | 51 | 113 | 311 |
| HDL-C | mmol/L | (-) | M | 18~65 | 287 | 0.60 | 0.76 | 0.68 | 1.16 | 1.83 | 1.73 | 1.93 | mg/dL | 26 | 45 | 71 |
| | | (-) | F | 18~65 | 333 | 0.67 | 0.78 | 0.72 | 1.30 | 2.02 | 1.92 | 2.12 | | 28 | 50 | 78 |
| LDL-C | mmol/L | (-) | MF | ~45 | 373 | 1.31 | 1.53 | 1.42 | 2.66 | 4.28 | 4.09 | 4.48 | mg/dL | 55 | 103 | 166 |
| | | (-) | MF | 45~ | 310 | 1.22 | 1.55 | 1.38 | 2.98 | 4.79 | 4.59 | 4.98 | | 54 | 115 | 185 |
| nonHDL | mmol/L | (-) | M | 18~65 | 290 | 1.69 | 2.07 | 1.88 | 3.58 | 5.69 | 5.43 | 5.95 | mg/dL | 73 | 139 | 220 |
| | | (-) | F | ~45 | 195 | 1.56 | 2.09 | 1.83 | 3.17 | 4.84 | 4.59 | 5.09 | | 71 | 123 | 187 |
| | | (-) | F | 45~ | 167 | 2.09 | 2.53 | 2.31 | 3.85 | 5.85 | 5.60 | 6.10 | | 89 | 149 | 226 |
| AST | U/L | (+) | M | 18~65 | 227 | 11.2 | 13.5 | 12 | 23 | 38 | 35.0 | 40.6 | U/L | 12 | 23 | 38 |
| | | (+) | F | 18~65 | 255 | 9.6 | 11.2 | 10 | 19 | 32 | 29.2 | 33.8 | | 10 | 19 | 32 |

*(Continued)*

**Table 3.** (Continued)

| | | | | | | 90%CI of LL | | RI in SI unit | | | 90%CI of UL | | | RI in conv. unit | | |
|---|---|---|---|---|---|---|---|---|---|---|---|---|---|---|---|---|
| ALT | U/L | (+) | M | 18~65 | 226 | 2.9 | 5.4 | 4 | 20 | 66 | 55.4 | 76.4 | U/L | 4 | 20 | 66 |
| | | (+) | F | ~45 | 155 | 2.4 | 4.7 | 4 | 12 | 30 | 26 | 34 | | 4 | 12 | 30 |
| | | (+) | F | 45~ | 122 | 2.1 | 3.7 | 3 | 13 | 43 | 37 | 49 | | 3 | 13 | 43 |
| LDH | U/L | (-) | MF | 18~65 | 613 | 82 | 99 | 91 | 181 | 282 | 271 | 294 | U/L | 91 | 181 | 282 |
| ALP | U/L | (-) | M | 18~65 | 285 | 41 | 48 | 45 | 76 | 131 | 124 | 139 | U/L | 45 | 76 | 131 |
| | | (-) | F | ~45 | 192 | 28 | 35 | 31 | 60 | 98 | 86 | 111 | | 31 | 60 | 98 |
| | | (+) | F | 45~ | 120 | 30 | 46 | 38 | 76 | 126 | 115 | 137 | | 38 | 76 | 126 |
| GGT | U/L | (-) | M | 18~65 | 288 | 13 | 16 | 14 | 32 | 72 | 65 | 80 | U/L | 14 | 32 | 72 |
| | | (+) | F | ~45 | 155 | 9 | 13 | 11 | 20 | 38 | 32 | 43 | | 11 | 20 | 38 |
| | | (+) | F | 45~ | 123 | 9 | 15 | 12 | 26 | 62 | 49 | 75 | | 12 | 26 | 62 |
| CK | U/L | (-) | M | 18~65 | 283 | 24 | 38 | 31 | 108 | 254 | 219 | 289 | U/L | 31 | 108 | 254 |
| | | (-) | F | ~45 | 196 | 31 | 39 | 35 | 74 | 151 | 133 | 169 | | 35 | 74 | 151 |
| | | (+) | F | 45~ | 124 | 24 | 40 | 32 | 80 | 173 | 136 | 210 | | 32 | 80 | 173 |
| AMY | U/L | (-) | M | 18~65 | 290 | 24 | 31 | 28 | 69 | 142 | 134 | 149 | U/L | 28 | 69 | 142 |
| | | (-) | F | 18~65 | 332 | 21 | 30 | 26 | 60 | 113 | 105 | 122 | | 26 | 60 | 113 |
| IgG | g/L | (+) | M | 18~65 | 199 | 7.68 | 8.60 | 8.1 | 12.5 | 17.3 | 16.56 | 17.94 | mg/dL | 814 | 1248 | 1725 |
| | | (-) | F | 18~65 | 306 | 8.41 | 9.10 | 8.8 | 13.1 | 19.4 | 18.59 | 20.19 | | 875 | 1311 | 1939 |
| IgA | g/L | (-) | MF | 18~65 | 539 | 0.95 | 1.16 | 1.06 | 2.22 | 4.43 | 4.18 | 4.67 | mg/dL | 106 | 222 | 443 |
| IgM | g/L | (-) | MF | 18~65 | 544 | 0.41 | 0.48 | 0.44 | 1.19 | 2.76 | 2.59 | 2.93 | mg/dL | 44 | 119 | 276 |
| C3* | g/L | (-) | MF | 18~65 | 546 | 0.94 | 1.05 | 1.00 | 1.57 | 2.38 | 2.29 | 2.47 | mg/dL | 100 | 157 | 238 |
| C4 | g/L | (-) | MF | 18~65 | 546 | 0.14 | 0.15 | 0.14 | 0.29 | 0.50 | 0.48 | 0.52 | mg/dL | 14 | 29 | 50 |
| CRP | mg/L | (+) | M | 18~65 | 200 | 0.0 | 0.8 | 0.4 | 2.4 | 25.2 | 13.5 | 36.8 | mg/dL | 0.04 | 0.24 | 2.52 |
| | | (-) | F | 18~65 | 309 | 0.3 | 0.7 | 0.5 | 2.7 | 35.3 | 25.7 | 44.9 | | 0.05 | 0.27 | 3.53 |
| PTH | ng/L | (-) | MF | 18~65 | 544 | 19 | 22 | 20 | 45 | 91 | 86 | 96 | pg/mL | 20 | 45 | 91 |

It is of note that in the derivation of RIs for lipids and PTH, possible influence of lipid lowering drugs and vitamin D and/or calcium were considered. However, the use of statin drugs was reported only by four volunteers, vitamin D by two, and calcium by one. Therefore, we ignored those external factors in the derivation of respective RIs.

We explored the difference in regression lines between BMI and RVs of six nutritional markers (LDL-C, HDL-C, ALT, GGT, CRP, and UA) across ten countries that collaborated in the global study. It is notable that regression lines showed different slopes, very steep in some countries (implying slight change induces profound effect on the RVs) while moderate to gentle in others. Taking HDL-C as an example, it was previously [16] reported that the HDL-C concentration did not reduce linearly as BMI advanced >30 kg/m$^2$. This could explain the wide differences in the regression line slopes among the different populations listed in **Fig 3**, and the tendency of the Middle Eastern populations (Turkey, Saudi Arabia, and Egypt) with higher BMI to have a weaker slope compared to other populations (Japan, China, India) with lower mean BMI. This finding may explain a relatively low prevalence of metabolic syndrome by NECP ATPIII criteria [17] among the participants (calculated as 20% of males and 16.1% of females) despite high obesity rates. However, the predominance of dyslipidemia revealed in this study is an alarming signal in consideration of high cardiovascular mortality in Egyptian [18].

Regarding regional difference in RVs, the comparison of DBil RIs indicated a lower DBil among southern Egyptians. Nevertheless, we could not derive a separate RI for this group due to the small number of this particular subset. This finding deserves further study to evaluate

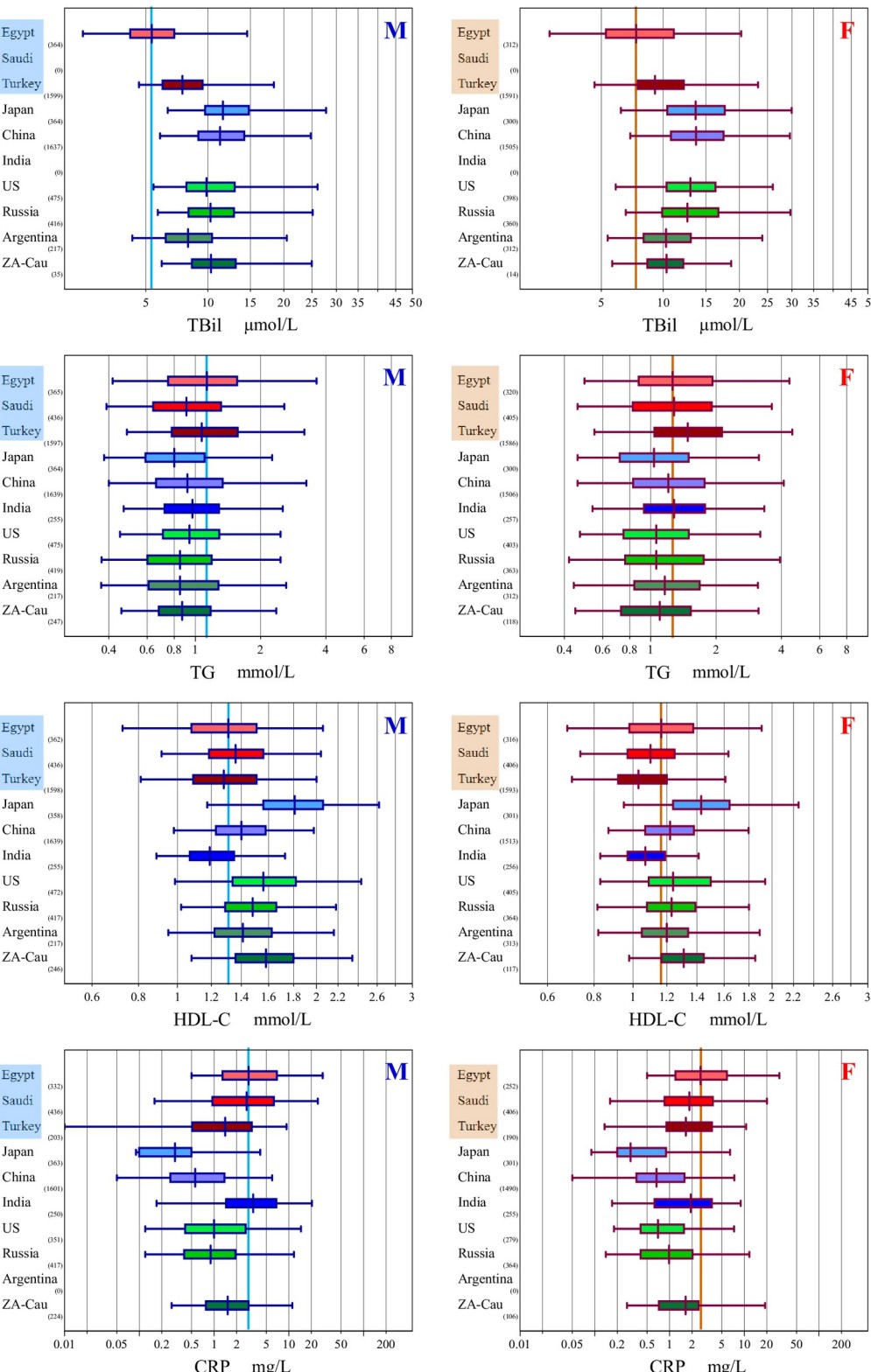

**Fig 2. Side-by-side comparison of RVs for TBil, TG, HDL-C, and CRP between Egyptian and other population.**
RVs of ten countries were compared by box-whisker charts. The box represents central 50% ranges, the vertical bar at the box represents the median. The span of the horizontal bar represents central 95% interval. The countries were subgroups by ethnicity: Middle Easterners: "Saudi Arabia, Turkey" with the central box shaded in red, Asians: "Japan, China, India" shaded in blue, and Caucasians: "USA, Russia, Argentina, South African Caucasians (ZA-Cau)" shades in green. The full-range vertical line represents median RV of Egyptian male (blue) and female (red).

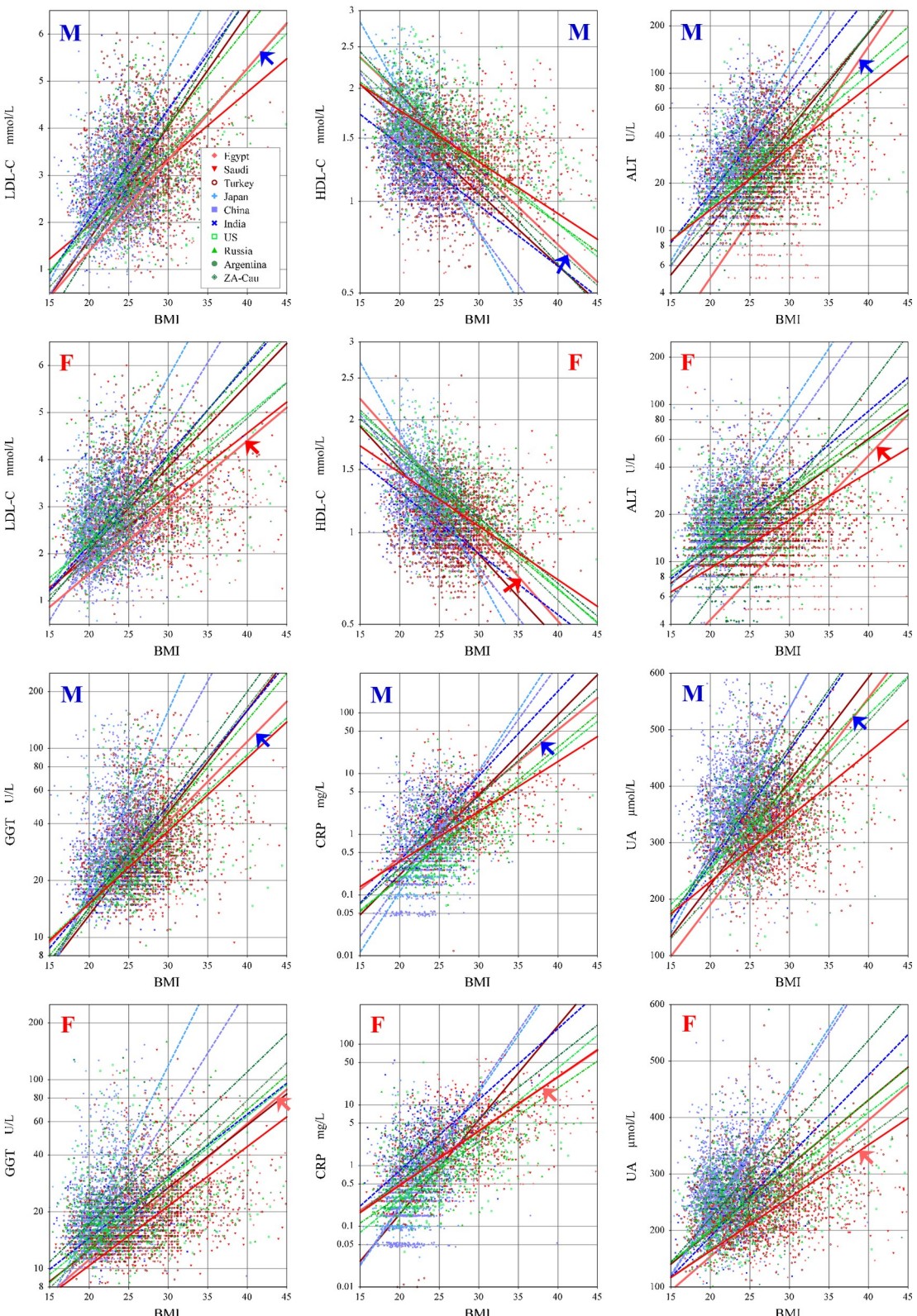

**Fig 3. Ethnic differences in regression line between BMI and RVs of nutritional markers.** Comparison of least-square linear regression lines between BMI and RVs of six nutritional markers among ten countries: Middle Easterners: "Egypt, Saudi Arabia, Turkey" in graded red, Asians: "Japan, China, India" in graded blue, and Caucasians "USA, Russia, Argentina, South African Caucasians (ZA-Cau)" in graded green.

whether it is due to a genetic variation in bilirubin conjugation pathway or a mere reflection of a different nutritional state.

Regarding the influence of smoking, the habit was self-reported by 27.5% of male and 1.1% of females. Previous reports are mixed in the direction of the relationship between smoking and IgA: increased in some studies [19,20], while reduced in others [21]. In our study, the IgA was positively correlated with smoking. Whereas, smoking habit-related lowering of IgG or HDL-C has been reported [9,22]. Such a finding was not observed in this study despite the appreciable number of male smokers.

## Conclusion

Standardized RIs for 34 chemistry analytes among Egyptians were established considering various sources of variations. They can be use in common in Egypt except a few analytes that showed moderate differences between northern and southern Egypt. Despite the high prevalence of obesity in Egyptians, the RVs of nutritional markers were found less sensitive to increased BMI, especially in females, when compared to other collaborating countries. From an international perspective, this study revealed that RVs of Egyptian feature low HDL-C and TBil, and high TG and CRP.

## Supporting information

**S1 Fig. Comparison of panel test results from two testing centers and assigned values.** Value-assigned panel of sera were tested in Cairo and Alexandria University and their test results were compared with the assigned values (11). The linear regression was computed by major-axis regression line. Merging of volunteers' test results were done by aligning them to the assigned values.
(PDF)

**S2 Fig. Sex and age-related changes in RVs of all parameters.** RVs were partitioned by sex (male:M, female: F) and age-subgroups (~29, 30~39, 40~49, 50~). The box in the center of each scattergram indicates the mid 50% range of RVs, and its central vertical bar represents the median. The data size is shown at the right bottom of the age group labels. Because no secondary exclusion was done for RVs, the range of the scatter plot may not match to the RI to be determined.
(PDF)

**S3 Fig. Between-region differences in RVs observed in five analytes.** RVs of five analytes, TBil, DBil, Glu, TG, and LDL-C that showed high SDR for between-region differences (SDRreg) were partitioned into four groups by sex and region (North vs. South Egypt). Values of BMI were also shown subgrouped by sex and region to prove that the regionality is independent of the levels of BMI.
(PDF)

**S1 Table. The assay methods and traceability with formulae for calculated parameters.**
(XLSX)

**S2 Table. Demographic profile of the participants.**
(XLSX)

**S3 Table. The list of all RIs partitioned by sex.**
(XLSX)

**S4 Table. The list of all RIs partitioned by sex and age.**
(XLSX)

**S1 Data.**
(XLSX)

## Author Contributions

**Conceptualization:** Heba Baz, Kiyoshi Ichihara, Ola Elgaddar.

**Data curation:** Heba Baz, Kiyoshi Ichihara, May Selim, Ola Elgaddar.

**Formal analysis:** Heba Baz, Kiyoshi Ichihara, Ahmed Awad, Sarah Aglan, Ola Elgaddar.

**Funding acquisition:** Kiyoshi Ichihara, May Selim, Ahmed Awad, Ola Elgaddar.

**Investigation:** Heba Baz, Kiyoshi Ichihara, May Selim, Ahmed Awad, Sarah Aglan, Ola Elgaddar.

**Methodology:** Heba Baz, Kiyoshi Ichihara, May Selim, Ahmed Awad, Sarah Aglan.

**Project administration:** Sarah Aglan, Ola Elgaddar.

**Resources:** Heba Baz, Ahmed Awad, Amina Hassab, Ola Elgaddar.

**Software:** Kiyoshi Ichihara.

**Supervision:** Heba Baz, Kiyoshi Ichihara, Dalia Ramadan, Amina Hassab, Lamia Mansour, Ola Elgaddar.

**Validation:** Kiyoshi Ichihara, Sarah Aglan.

**Visualization:** Dalia Ramadan.

**Writing – original draft:** Heba Baz, Kiyoshi Ichihara.

**Writing – review & editing:** Heba Baz, Kiyoshi Ichihara, Lamia Mansour.

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
