## [Decision Letter · Decision Letter 0]

18 Sep 2020

PONE-D-20-21548

Establishment of Reference Intervals of Clinical Chemistry Analytes for the Adult Population in Egypt

PLOS ONE

Dear Dr. Ichihara,

Thank you for submitting your manuscript to PLOS ONE. After careful consideration, we feel that it has merit but does not fully meet PLOS ONE’s publication criteria as it currently stands. Therefore, we invite you to submit a revised version of the manuscript that addresses the points raised during the review process. A response to reviewers should point out the page and what you change in the manuscript.

We look forward to receiving your revised manuscript.

Kind regards,

Nguyen Tien Huy, Ph.D., M.D.

Academic Editor

PLOS ONE

Journal Requirements:

Reviewers' comments:

Reviewer's Responses to Questions

**Comments to the Author**

1. Is the manuscript technically sound, and do the data support the conclusions?

Reviewer #1: Yes

Reviewer #2: Yes

2. Has the statistical analysis been performed appropriately and rigorously? 

Reviewer #1: Yes

Reviewer #2: Yes

3. Have the authors made all data underlying the findings in their manuscript fully available?

Reviewer #1: Yes

Reviewer #2: Yes

4. Is the manuscript presented in an intelligible fashion and written in standard English?

Reviewer #1: Yes

Reviewer #2: Yes

5. Review Comments to the Author

Reviewer #1: First the topic is so interesting and the benefits derived from it would be so important in the future.

1/ adult population includes those >65 years but this age class was some kind of ''ignored'' throughout the whole study, not even age and gender equalization was done for this class justify this please ?

2/ Are areas of origin of participant representative of all the country ?

3/ Was a sample size evaluation done before conducting the study ?

4/ The health status questionnaire provided for participant please provide a copy of it so that we can evaluate the sample

5/ So only two laboratories agreed to participate or were they the only ''reference'' lab , else how were participants divided between them ?

6/ Suppl table 3 and table 2 no age>65 y provided

7/ Suppl table 3 why stratifying by age 45y ?

8/ Method for calculating eGFR, aCa, TIBC, HDL were not provided

9/ assay timeline for samples collected of 2-3 months and 8 months reference for that please ?

10/ Dividing Egypt into North and South was done according to what reference ?

11/ Please mention clearly inclusion and exclusion criteria of participants as no criteria provided

12/ ''did not deny alcohol drinking'' and low smoking status (mainly female++)  Response Bias suspected

else what about other drugs consumption ? was a thorough history of patients done before with the questionnaire?

13/ Supp fig1 ''Asn val'' stands for what ?

14/ Discussion

Not well elaborated and lacks many referencing

*I was expecting a reference on the prevalence of obesity in Egypt

*i was expecting a reference on the prevalence of smoking, alcohol drinking, drugs consumption, and other habits in the population so that was evaluate if the sample results are possible to generalize to the whole population

* bilirub level reduced maybe a reference on the prevalence of cholecystitis or cholangitis in Egypt

* no relation between IgG and HDL provided a possible explanation or a reference for that

15/ NO stating of NEW results in the discussion section please like fig3 or the prevalence of metabolic syndrome (which by the way suppose you calculated BP and wrist circumference of the sample ??)

-- all results should be first mentioned in the results section

Reviewer #2: please check below or the attached pdf file:

Editor in Chief, PLOS ONE

Establishment of Reference Intervals of Clinical Chemistry Analytes for the Adult

Population in Egypt

PONE-D-20-21548

Reviewer(s)' Comments to Author:

Reviewer:

Comments to the Author

The manuscript established the reference interval for 34 major chemistry analytes and discussed the sources of variation. The study is part of the global reference interval study of the IFCC.

The manuscript has many details about data analysis but some core aspects have weakened the manuscript which need to be answered and corrected.

Specific comments

Manuscript:

Material and methods

1. Study design

- The study was performed in two different institutes. Therefore, the ethical approval number should be unified but it seems Cairo University ethical approval number only has been mentioned. What about Medical Research Institute of Alexandria University was there any specific ethical approval number? If yes then please mention it.

2. Recruitment of reference individuals

“A total of 691 apparently healthy Egyptian volunteers were recruited from Cairo, Alexandria, Delta, Suez Canal, and Upper Egypt; Fayoum, Beni-Sueif or areas southern as faras Nubia”. It is not so clear if samples were collected from volunteers from each of these locations or volunteers came to one cite for samples to be collected.

4. Analytes and measurements

- Page 2: Suppl table 1: does not show the formulae for the calculated parameters

Globulins (Glb), estimated glomerular filtration rate (eGFR), adjusted calcium (aCa), total iron binding capacity (TIBC), and non-high-density lipoprotein (nonHDL).

- PTH is the only anlyte based on immunoassay measurement. Why it has been included in this study in particular while the study is dedicated to chemistry tests?

- PTH can be affected by calcium and vitamin D intake, did the author consider them when RI for PTH has been derived (as exclusion criteria). The same thing for Iron. If yes, then how many subjects were excluded?

- When the study was started (month/year) and when it was ended (month/year)?

- Page 3: IgG, IgM, IgA, C3, C4, CRP and PTH were measured 8 months after the collection, why and how their stability was assessed?

5. Between-center method comparison and standardization

- A panel composed of 50 healthy volunteers' sera with assigned values for 34 chemistry and immunoturbidimetry analytes was measured by the two different centers for merging and standardization of test results. How this panel of samples was transferred physically from one center to another? And how their stability was guaranteed?

- A summarized table is needed to show number of samples, BMI, sex distribution, age, BP and other anthropometric measurements for each cite and in total.

- In this study, samples were analyzed in two independent labs. The panel was used to standardize test results between both cites. In other involved countries in the global study all samples were sent to the main lab for analysis and a panel was used to standardize results between different labs. Clarify if the protocol followed in this study contradicts the harmonized protocol created by the C-RIDL or not.

- For between-day quality control, no data are shown.

Results:

- Page 6: The head title in Table 1 “South Egy” to be changes to “Region”.

- Table 3: TBil and DBil unit mmol/L to be corrected

TP, Alb, Glb unit g/L to be corrected in RI in conv, unit.

- Page 9: Supp Fig 3 shows low glucose, DBil and TBil for samples from South Egypt. In addition to this in Figure 2, TBil is the lowest compared to other countries (Glu is not included in the figure). It seems these outcomes are not compatible with the prevalence and rates of diabetes and hepatitis in Egypt compared to other countries.

Did the author investigate the possibility of the presence of any gap in the followed C-RIDL protocol regarding samples integrity i.e. during samples collection, transportation, delay in separation, sun exposure etc. before samples were analysed.

Discussion:

- Page 12: “Mean BMI in our study was 28.2±5.0” the unit needs to be added

6. PLOS authors have the option to publish the peer review history of their article (what does this mean?). If published, this will include your full peer review and any attached files.

Reviewer #1: **Yes: **Nacir Dhouibi

Reviewer #2: No

---

## [Author Response · Author response to Decision Letter 0]

26 Oct 2020

Reviewers' comments:

Reviewer's Responses to Questions

Comments to the Author

Reviewer #1: First the topic is so interesting and the benefits derived from it would be so important in the future.

Our response→ We are grateful for the detailed review of our paper and offer of invaluable comments for improvement.

1/ adult population includes those >65 years but this age class was some kind of ''ignored'' throughout the whole study, not even age and gender equalization was done for this class justify this please ?

Our response→ The C-RIDL’s harmonized protocol recommended the adult age range be set to 18~65 with flat age and sex distribution. However, for subjects above 65 years were requested to be included for use in elucidating the age-related changes of reference values in the high age ranges by pooling the results. Therefore, we did not include the results in our analysis except for determination of RIs for subjects above 65 years of age.

In the revised manuscript, we make clear of this point in the Method as below 

“Subjects were selected so that at least 80% were from the age of 18 to 65 years, with equal gender mix and age distributions, except for individuals over 65 years of age. Test results of the high age groups were not included in the determination of RIs for the main age group (18~65), but included in the derivation of RIs for the age group ≥65 years. The results of the higher age group were also meant for use by C-RIDL in elucidating age-related change profiles of RVs in a wide range by merging all results worldwide.” 

2/ Are areas of origin of participant representative of all the country ? 

Our response→ Yes, the regions included in the study represents Egypt, Nile valley, along with coastal cities which are the inhabited regions of Egypt. To make clear of the distribution of the regions, in the revised article we gave more clearer description as follows: “A total of 691 apparently healthy Egyptian volunteers were recruited from Cairo region [Cairo, Giza, Helwan], Alexandria, the Nile Delta [Beheyra, Daqahlia, Damietta, Gharbia, Kar Elshiekh, Menoufia], South (Upper) Egypt [Fayoum, Beni-Sueif , Minya, Assiut, Qena, and Nubia], and the Suez Canal [Port Said and Suez]”.

3/ Was a sample size evaluation done before conducting the study ?

Our response→ In the C-RIDL’s harmonized protocol, the rationale for the sample size is described as follows:

“The target sample size of practically attainable level from each country is set at minimum 500 or more, greater than twice the minimum number of 120×2 (male and female) recommended by C28-A3, so that country-specific RIs can be obtained in a more reproducible manner. The number is at least enough to make between-country comparisons of test results with a power of detecting SD ratio of 0.25 (SD due to a given source of variation relative to between-individual SD), allowing errors of α<0.05 and β<0.2, in statistical hypothesis testing of difference in means between any two countries done separately for each gender.”.

We targeted the sample size of ≥680 that exceeds the suggested minimum of 500 (both gender combined).

4/ The health status questionnaire provided for participant please provide a copy of it so that we can evaluate the sample

Our response→ Following the suggestion, we chose to include the questionnaire in Arabic and English as a supplemental material in the revised manuscript.

5/ So only two laboratories agreed to participate or were they the only ''reference'' lab , else how were participants divided between them ?

Our response→ The two labs agreed to act as “reference lab”. The samples were divided as follow: Cairo University recruited along the Nile river, from the Nile delta north to Nubia in the south. Alexandria University covered the coastal region along the Mediterranean. Both labs recruited from the Suez Canal region. However, we did not include this detailed information in the revised text.

6/ Suppl table 3 and table 2 no age>65 y provided

Our response→ As we described above for “1/” we just included test results of subjects≥65 years for derivation of RIs for the age group≥45 years, and thus they were not ignored in the Suppl Table 3.

7/ Suppl table 3 why stratifying by age 45y ?

Our response→ Judged from the C-RIDL interim report that demonstrated age-related change profile of major analytes [https://ars.els-cdn.com/content/image/1-s2.0-S0009898116303898-mmc4.pdf], most of the age-related changes occur in females starting from 40 years until 60 years. A candidate for demarcation point of year was either 45 or 50 year of age. We arbitrarily set the demarcation point as 45 years in order to get a higher precision in determining RIs: i.e., the number of females above 50 (n=117) was much smaller than those above 45 (n=165).We commented on this reasoning in the Methods of the revised manuscript as “The partitioning of RVs by age was done arbitrarily at 45 years to ensure enough data size for the higher age group ”.

8/ Method for calculating eGFR, aCa, TIBC, HDL were not provided

Our response→ We appreciate for pointing out the problem of missing formulae for the calculating parameters, which were inadvertently missed out in Suppl Table 1. We added the formulae in the bottom of the table.

9/ assay timeline for samples collected of 2-3 months and 8 months reference for that please ?

Our response→ The difference in the period of storage before measurements (2-3 months or general chemistry, and 8 months for immunochemistry) was simply caused by availability of support for assay reagents. If this query was made for a concern on the stability of serum specimens stored at −80ºC, please refer to our responses to Reviewer-2’s query on sample stability below. 

10/ Dividing Egypt into North and South was done according to what reference ?

Our response→ We realized that we did not give explanation on the distinction of regions. Egypt is split historically and geographically into, North (Lower) Egypt comprised of the Nile Delta Northern to Cairo reaching Mediterranean, and South (Upper) Egypt comprised of cities along the Nile river banks southern to Cairo reaching the Sudanese borders. In this paper, we used the terminology of North and South Egypt instead of the Lower and Upper Egypt conventionally used in Egypt. Accordingly we added the following description in the Methods: “Dividing the Egyptian population to north and south was adopted in respect to demographic and social differences between the two regions. Osman et al., 2016.”

Reference: https://assets.publishing.service.gov.uk/government/uploads/system/uploads/attachment_data/file/793089/Egypt_Toponymic_Factfile-March_2019.pdf

11/ Please mention clearly inclusion and exclusion criteria of participants as no criteria provided

Our response→ Following the suggestion, we added the inclusion/exclusion criteria in the revised Methods as follows: “We followed the inclusion and exclusion criteria of C-RIDL protocol: Participants of the study were included if they are feeling subjectively well, older than 18 years with no upper limit of age for participation.　Those who take less equal 3 drugs for minor conditions or nutritional supplements were allowed.. All the following were excluded; participants who were known diabetics on insulin or oral therapy, , who had a positive history of a hepatic or renal disease, who had an grossly abnormal test results in the previous year, who were hospitalized within the previous 4 weeks prior to joining the study, who donated blood in the previous 3 months, known carrier for HCV, HBV or HIV, who participated recently in a clinical trial for an investigational product.” 

12/ ''did not deny alcohol drinking'' and low smoking status (mainly female++)  Response Bias suspected else what about other drugs consumption ? was a thorough history of patients done before with the questionnaire?

Our response→ We understand the expression sounds odd, however, 90% of the Egypt’s population are of Muslim faith. Although alcohol is available for purchase, yet, its consumption is frowned upon in the public eyes, thus the participants who candidly admitted to its use in this study were minimum. According to Rabie et al., 2020, the prevalence of alcohol use in the past 12 months is 2.9% of the population. Therefore, we believe there is no response bias in the status of alcohol drink. In any case, we added a supplemental table showing the demographic information of the volunteers including habits of smoking and drinking alcohol as Suppl Table 2.

With regards to drugs, we added this information in the revised Results. “from the health status questionnaire, we found regular use of drugs in the following frequencies: anti-hypertensives in 13 volunteers (1.9%), analgesics 51 (7.5%), anti-allergic 10 (1.6%), antacid 24 (3.5%), statin 4 (0.6%), others 8 (1.2%). We disregard the information in the subsequent analyses because they were all minor and small in dosage”. 

13/ Supp fig1 ''Asn val'' stands for what ?

Our response→ We realized the problem of missing description for the abbreviation. We added it in the footnote of the figure as “assigned values”.

14/ Discussion

Not well elaborated and lacks many referencing

*I was expecting a reference on the prevalence of obesity in Egypt

*i was expecting a reference on the prevalence of smoking, alcohol drinking, drugs consumption, and other habits in the population so that was evaluate if the sample results are possible to generalize to the whole population

* bilirub level reduced maybe a reference on the prevalence of cholecystitis or cholangitis in Egypt

* no relation between IgG and HDL provided a possible explanation or a reference for that

Our response→ The WHO has reported a significantly high prevalence of obesity in Egypt among six other Arab world countries, ranging from 74% to 86% in women and 69% to 77% in men. Also, Egypt is suffering from high tobacco burden, 40.5% of men, 0.3% of women, and 20.3% of Egypt’s population overall are daily tobacco smokers. We understand that the prevalence of cholecystitis and cholangitis is irrelevant to the levels of bilirubin in healthy individuals, and thus, we could not add any reference on its prevalence in Egypt in comparison to other countries. In any case, please note that we added Suppl Table 2 showing the demographic information regarding BMI, smoking and drinking habits.

15/ NO stating of NEW results in the discussion section please like fig3 or the prevalence of metabolic syndrome (which by the way suppose you calculated BP and wrist circumference of the sample ??)

-- all results should be first mentioned in the results section

Our response→ We appreciate for pointing out the problem for us. We moved the explanation on Fig.3 to the end of the Results section under the subheading of “Comparison of Egyptian BMI-related changes with other populations”.

Reviewer #2: please check below or the attached pdf file:

Comments to the Author

The manuscript established the reference interval for 34 major chemistry analytes and discussed the sources of variation. The study is part of the global reference interval study of the IFCC.

The manuscript has many details about data analysis but some core aspects have weakened the manuscript which need to be answered and corrected.

Our response→ We are grateful for the critical review of our paper and the kind offer of invaluable comments to improve it.

Specific comments

Manuscript:

Material and methods

1. Study design

- The study was performed in two different institutes. Therefore, the ethical approval number should be unified, but it seems Cairo University ethical approval number only has been mentioned. What about Medical Research Institute of Alexandria University was there any specific ethical approval number? If yes then please mention it.

Our response→ The study protocol was approved by Cairo University and the protocol included collecting cases from several governorates and performing testing in more than one lab (as per the geographical distribution), and since we were using the exact same protocol in all locations we used only one ethical approval which is that of Cairo University.

2. Recruitment of reference individuals

“A total of 691 apparently healthy Egyptian volunteers were recruited from Cairo, Alexandria, Delta, Suez Canal, and Upper Egypt; Fayoum, Beni-Sueif or areas southern as faras Nubia”. It is not so clear if samples were collected from volunteers from each of these locations or volunteers came to one cite for samples to be collected.

Our response→ In the revised text, we made it clear where the samples were collected in response to the comment from Reviewer-1. The sampling were done at each location and specimens were brought to either of the two central labs located in Cairo and Alexandria University. 

4. Analytes and measurements

- Page 2: Suppl table 1: does not show the formulae for the calculated parameters

Globulins (Glb), estimated glomerular filtration rate (eGFR), adjusted calcium (aCa), total iron binding capacity (TIBC), and non-high-density lipoprotein (nonHDL).

Our response→ We appreciate for pointing out the problem of missing formulae for the calculating parameters, which were inadvertently missed out in Suppl Table 1. We added the formulae in the table.

- PTH is the only anlyte based on immunoassay measurement. Why it has been included in this study in particular while the study is dedicated to chemistry tests?

Our response→ The reason why we included just PTH from among many analytes measured by immunoassays was simply a matter of limited budget to acquire reagents.

- PTH can be affected by calcium and vitamin D intake, did the author consider them when RI for PTH has been derived (as exclusion criteria). The same thing for Iron. If yes, then how many subjects were excluded?

Our response→ From the health-status questionnaire, we obtained information on regular use of supplements and medications from all volunteers. However, there were only 3 individuals with use of vitamin D and/or calcium. Therefore, we regarded the serum levels of PTH were not affected by the external factor. We wrote about this point in the Results as follows: “It is of note that in the derivation of RIs for lipids and PTH, possible influence of lipid lowering drugs and vitamin D and/or calcium were considered. However, the use of statin drugs was reported only by four volunteers, vitamin D by two, and calcium by one. Therefore, we ignored those external factors in the derivation of respective RIs.”

- When the study was started (month/year) and when it was ended (month/year)?

Our response→ Study was approved to start on January 31st, 2015, and the final data analysis ended early 2020. We added this study period in the revised manuscript as “The study was approved by the ethical committee of Cairo University (N-13-2015) on January 31st, 2015, and the final data analysis ended early 2020.”

- Page 3: IgG, IgM, IgA, C3, C4, CRP and PTH were measured 8 months after the collection, why and how their stability was assessed?

Our response→ With difficulty of recruiting sufficient number of volunteers from many parts of Egypt, we were obliged to take many months to achieve our target sample size of >650. Therefore, as described in the C-RIDL harmonized protocol [doi:10.1515/cclm-2013-0249], we stored all serum specimens at −80ºC until the time of collective measurement. The stability of serum specimens stored at −80ºC has been generally assumed in the protocol. However, we confirmed it as described just below, including the stability even for C3, C4, PTH, and insulin, which are known to be unstable at room temperature and regular freezing temperature.

5. Between-center method comparison and standardization

- A panel composed of 50 healthy volunteers' sera with assigned values for 34 chemistry and immunoturbidimetry analytes was measured by the two different centers for merging and standardization of test results. How this panel of samples was transferred physically from one center to another? And how their stability was guaranteed?

Our response→ Two sets of the serum panel (a second lot produced in March 2014 in Japan) was transported to Cairo in 2015, packed in dry ice. Transportation within Egypt of the one for Alexandria was also done packed in dry ice. The stability of almost any analyte in serum specimens if stored at −80ºC is well documented [doi:10.1016/j.clinbiochem.2012.03.029]. 

However, to confirm the finding, we evaluated the reproducibility of panel test results measured 1 to 4 years after storage at −80ºC using the same analyzer. We could prove a perfect stability of 29 major chemistry analytes except LDH (S-Fig. 1), and of 6 representative analytes measured by immunoassays including PTH (S-Fig.2). Although test results of LDH seem lowered in the second testing, it turned out to be due to between-day bias of measurement, judged from S-Fig. 3, which demonstrates fluctuation of panel test results (n=50) measured at 14 labs over the 5 year period (2014~2019) including Egypt.

Please allow us not to include S-Fig. 1 to 3 in this paper, because we plan to report the findings in a different paper. 

- A summarized table is needed to show number of samples, BMI, sex distribution, age, BP and other anthropometric measurements for each cite and in total.

Our response→ Following the suggestion. We newly made a table (Suppl Table 2) showing those demographic information. Therefore, the original Suppl Table 2 was renumbered accordingly.

- In this study, samples were analyzed in two independent labs. The panel was used to standardize test results between both cites. In other involved countries in the global study all samples were sent to the main lab for analysis and a panel was used to standardize results between different labs. Clarify if the protocol followed in this study contradicts the harmonized protocol created by the C-RIDL or not.

Our response→ We understand we followed the protocol faithfully. The test results of the two central labs were merged based on the panel test results. For the standardized analytes, test results of the two labs were first aligned to the assigned values and then merged. Whereas, for non-standardizable analytes test results were merged to the values of Cairo University by use of major axis regression line: (Cairo-value) = a + b×(Alex-value).

- For between-day quality control, no data are shown.

Our response→ According to the suggestion, we added between-day CVs to the Suppl Table 1 that lists names and abbreviations of all analytes and analytical methods.

Results:

- Page 6: The head title in Table 1 “South Egy” to be changes to “Region”.

Our response→ We changed the notation accordingly. 

- Table 3: TBil and DBil unit mmol/L to be corrected

TP, Alb, Glb unit g/L to be corrected in RI in conv, unit.

Our response→ We appreciate for letting us know of the problem.

- Page 9: Supp Fig 3 shows low glucose, DBil and TBil for samples from South Egypt. In addition to this in Figure 2, TBil is the lowest compared to other countries (Glu is not included in the figure). It seems these outcomes are not compatible with the prevalence and rates of diabetes and hepatitis in Egypt compared to other countries.

Did the author investigate the possibility of the presence of any gap in the followed C-RIDL protocol regarding samples integrity i.e. during samples collection, transportation, delay in separation, sun exposure etc. before samples were analysed.

Our response→ For the finding in Supp Fig 3 of low DBil and TBil in South Egypt, we postulate that it may be attributed to a genetic factor with no pertinent information to explain the finding. However, we do not think it was caused by a difference in the prevalence of hepatitis in the two regions (North vs. South). In fact, we denied the presence of any regional difference in the levels of AST and ALT. For the finding of low Egyptian TBil, we again assume it due to genetic/ethnic factor with a low level of TBil in Turkey. 

Although we did not show the figures comparing Glu levels with other countries, we did not find any difference in the levels of Egyptian Glu. The low levels of Glu in females of South Egypt, we do not think it due to a problem in sample processing, because such a finding was not apparent in males. We assume it due to less intake of carbohydrate in females of the South. In any case, we followed the C-RIDL protocol to ensured sample integrity at all times, and thus we do not think of any problem in preanalytical processing. 

Discussion:

- Page 12: “Mean BMI in our study was 28.2±5.0” the unit needs to be added

Our response→ We appreciate for letting us know of the problem. We added the unit of kg/m2.

---

## [Decision Letter · Decision Letter 1]

10 Dec 2020

Establishment of Reference Intervals of Clinical Chemistry Analytes for the Adult Population in Egypt

PONE-D-20-21548R1

Dear Dr. Ichihara,

We’re pleased to inform you that your manuscript has been judged scientifically suitable for publication and will be formally accepted for publication once it meets all outstanding technical requirements.

Kind regards,

Colin Johnson, Ph.D.

Academic Editor

PLOS ONE

Additional Editor Comments (optional):

Reviewers' comments:

Reviewer's Responses to Questions

**Comments to the Author**

1. If the authors have adequately addressed your comments raised in a previous round of review and you feel that this manuscript is now acceptable for publication, you may indicate that here to bypass the “Comments to the Author” section, enter your conflict of interest statement in the “Confidential to Editor” section, and submit your "Accept" recommendation.

Reviewer #1: All comments have been addressed

Reviewer #2: All comments have been addressed

2. Is the manuscript technically sound, and do the data support the conclusions?

Reviewer #1: Yes

Reviewer #2: Yes

3. Has the statistical analysis been performed appropriately and rigorously? 

Reviewer #1: I Don't Know

Reviewer #2: Yes

4. Have the authors made all data underlying the findings in their manuscript fully available?

Reviewer #1: Yes

Reviewer #2: Yes

5. Is the manuscript presented in an intelligible fashion and written in standard English?

Reviewer #1: Yes

Reviewer #2: Yes

6. Review Comments to the Author

Reviewer #1: (No Response)

Reviewer #2: This manuscript represents the conclusion of authors' hard work. Thanks for addressing all points as your responses were acceptable and satisfied.

7. PLOS authors have the option to publish the peer review history of their article (what does this mean?). If published, this will include your full peer review and any attached files.

Reviewer #1: **Yes: **Nacir Dhouibi

Reviewer #2: **Yes: **Dr Anwar Borai, King Abdullah International Medical Research Center, King Saud bin Abdulaziz University for Health Sciences, Pathology, King Abdulaziz Medical City, Jeddah, Saudi Arabia.

---

## [Editor Report · Acceptance letter]

23 Dec 2020

PONE-D-20-21548R1 

Establishment of Reference Intervals of Clinical Chemistry Analytes for the Adult Population in Egypt 

Dear Dr. Ichihara:

I'm pleased to inform you that your manuscript has been deemed suitable for publication in PLOS ONE. Congratulations! Your manuscript is now with our production department. 

Kind regards, 

on behalf of

Dr. Colin Johnson 

Academic Editor

PLOS ONE